# Adversarial Regularizers in Inverse Problems

**Sebastian Lunz**
DAMTP
University of Cambridge
Cambridge CB3 0WA
lunz@math.cam.ac.uk

**Ozan Öktem**
Department of Mathematics
KTH - Royal Institute of Technology
100 44 Stockholm
ozan@kth.se

**Carola-Bibiane Schönlieb**
DAMTP
University of Cambridge
Cambridge CB3 0WA
cbs31@cam.ac.uk

## Abstract

Inverse Problems in medical imaging and computer vision are traditionally solved using purely model-based methods. Among those variational regularization models are one of the most popular approaches. We propose a new framework for applying data-driven approaches to inverse problems, using a neural network as a regularization functional. The network learns to discriminate between the distribution of ground truth images and the distribution of unregularized reconstructions. Once trained, the network is applied to the inverse problem by solving the corresponding variational problem. Unlike other data-based approaches for inverse problems, the algorithm can be applied even if only unsupervised training data is available. Experiments demonstrate the potential of the framework for denoising on the BSDS dataset and for computed tomography reconstruction on the LIDC dataset.

## 1   Introduction

Inverse problems naturally occur in many applications in computer vision and medical imaging. A successful classical approach relies on the concept of variational regularization [11, 24]. It combines knowledge about how data is generated in the forward operator with a regularization functional that encodes prior knowledge about the image to be reconstructed.

The success of neural networks in many computer vision tasks has motivated attempts at using deep learning to achieve better performance in solving inverse problems [15, 2, 25]. A major difficulty is the efficient usage of knowledge about the forward operator and noise model in such data driven approaches, avoiding the necessity to relearn the physical model structure.

The framework considered here aims to solve this by using neural networks as part of variational regularization, replacing the typically hand-crafted regularization functional with a neural network. As classical learning methods for regularization functionals do not scale to the high dimensional parameter spaces needed for neural networks, we propose a new training algorithm for regularization functionals. It is based on the ideas in Wasserstein generative adversarial models [5], training the network as a critic to tell apart ground truth images from unregularized reconstructions.

Our contributions are as follows:

1. We introduce the idea of learning a regularization functional given by a neural network, combining the advantages of the variational formulation for inverse problems with data-driven approaches.

2. We propose a training algorithm for regularization functionals which scales to high dimensional parameter spaces.

3. We show desirable theoretical properties of the regularization functionals obtained this way.

4. We demonstrate the performance of the algorithm for denoising and computed tomography.

## 2 Background

### 2.1 Inverse Problems in Imaging

Let $X$ and $Y$ be reflexive Banach spaces. In a generic inverse problem in imaging, the image $x \in X$ is recovered from measurement $y \in Y$, where

$$y = Ax + e. \tag{1}$$

$A : X \to Y$ denotes the linear forward operator and $e \in Y$ is a random noise term. Typical tasks in computer vision that can be phrased as inverse problems include denoising, where $A$ is the identity operator, or inpainting, where $A$ is given by a projection operator onto the complement of the inpainting domain. In medical imaging, common forward operators are the Fourier transform in magnetic resonance imaging (MRI) and the ray transform in computed tomography (CT).

### 2.2 Deep Learning in Inverse Problems

One approach to solve (1) using deep learning is to directly learn the mapping $y \to x$ using a neural network. While this has been observed to work well for denoising and inpainting [28], the approach can become infeasible in inverse problems involving forward operator with a more complicated structure [4] and when only very limited training data is available. This is typically the case in applications in medical imaging.

Other approaches have been developed to tackle inverse problems with complex forward operators. In [15] an algorithm has been suggested that first applies a pseudo-inverse to the operator $A$, leading to a noisy reconstruction. This result is then denoised using deep learning techniques. Other approaches [1, 14, 25] propose applying a neural network iteratively. Learning proximal operators for solving inverse problems is a further direction of research [2, 19].

### 2.3 Variational regularization

Variational regularization is a well-established model-based method for solving inverse problems. Given a single measurement $y$, the image $x$ is recovered by solving

$$\operatorname{argmin}_x \|Ax - y\|_2^2 + \lambda f(x), \tag{2}$$

where the data term $\|Ax - y\|_2^2$ ensures consistency of the reconstruction $x$ with the measurement $y$ and the regularization functional $f : X \to \mathbb{R}$ allows us to insert prior knowledge onto the solution $x$. The functional $f$ is usually hand-crafted, with typical choices including total variation (TV) [23] which leads to piecewise constant images and total generalized variation (TGV) [16], generating piecewise linear images.

## 3 Learning a regularization functional

In this paper, we design a regularization functional based on training data. We fix a-priori a class of admissible regularization functionals $\mathcal{F}$ and then learn the choice $\{f\}_{f \in \mathcal{F}}$ from data. Existing approaches to learning a regularization functionals are based on the idea that $f$ should be chosen such that a solution to the variational problem

$$\operatorname{argmin}_x \|Ax - y\|_2^2 + \lambda f(x), \tag{3}$$

best approximates the true solution. Given training samples $(x_j, y_j)$, identifying $f$ using this method requires one to solve the bilevel optimization problem [17, 9]

$$\operatorname{argmin}_{f \in \mathcal{F}} \sum_j \|\tilde{x}_j - x_j\|_2, \quad \text{subject to} \quad \tilde{x}_j \in \operatorname{argmin}_x \|Ax - y_j\|_2^2 + f(x). \tag{4}$$

But this is computationally feasible only for small sets of admissible functions $\mathcal{F}$. In particular, it does not scale to sets $\mathcal{F}$ parametrized by some high dimensional space $\Theta$.

We hence apply a novel technique for learning the regularization functional $f \in \mathcal{F}$ that scales to high dimensional parameter spaces. It is based on the idea of learning to discriminate between noisy and ground truth images.

In particular, we consider approaches where the regularization functional is given by a neural network $\Psi_\Theta$ with network parameters $\Theta$. In this setting, the class $\mathcal{F}$ is given by the functions that can be parametrized by the network architecture of $\Psi$ for some choice of parameters $\Theta$. Once $\Theta$ is fixed, the inverse problem (1) is solved by

$$\text{argmin}_x \|Ax - y\|_2^2 + \lambda \Psi_\Theta(x). \tag{5}$$

## 3.1 Regularization functionals as critics

Denote by $x_i \in X$ independent samples from the distribution of ground truth images $\mathbb{P}_r$ and by $y_i \in Y$ independent samples from the distribution of measurements $\mathbb{P}_Y$. Note that we only use samples from both marginals of the joint distribution $\mathbb{P}_{X \times Y}$ of images and measurement, i.e. we are in the setting of unsupervised learning.

The distribution $\mathbb{P}_Y$ on measurement space can be mapped to a distribution on image space by applying a -potentially regularized- pseudo-inverse $A_\delta^\dagger$. In [15] it has been shown that such an inverse can in fact be computed efficiently for a large class of forward operators. This in particular includes Fourier and ray transforms occurring in MRI and CT. Let

$$\mathbb{P}_n = (A_\delta^\dagger)_\# \mathbb{P}_Y$$

be the distribution obtained this way. Here, $\#$ denotes the push-forward of measures, i.e. $A_\delta^\dagger Y \sim (A_\delta^\dagger)_\# \mathbb{P}_Y$ for $Y \sim \mathbb{P}_Y$. Samples drawn from $\mathbb{P}_n$ will be corrupted with noise that both depends on the noise model $e$ as well as on the operator $A$.

A good regularization functional $\Psi_\Theta$ is able to tell apart the distributions $\mathbb{P}_r$ and $\mathbb{P}_n$- taking high values on typical samples of $\mathbb{P}_n$ and low values on typical samples of $\mathbb{P}_r$ [7]. It is thus clear that

$$\mathbb{E}_{X \sim \mathbb{P}_r} [\Psi_\Theta(X)] - \mathbb{E}_{X \sim \mathbb{P}_n} [\Psi_\Theta(X)]$$

being small is desirable. With this in mind, we choose the loss functional for learning the regularizer to be

$$\mathbb{E}_{X \sim \mathbb{P}_r} [\Psi_\Theta(X)] - \mathbb{E}_{X \sim \mathbb{P}_n} [\Psi_\Theta(X)] + \lambda \cdot \mathbb{E} \left[ (\|\nabla_x \Psi_\Theta(X)\| - 1)_+^2 \right]. \tag{6}$$

The last term in the loss functional serves to enforce the trained network $\Psi_\Theta$ to be Lipschitz continuous with constant one [13]. The expected value in this term is taken over all lines connecting samples in $\mathbb{P}_n$ and $\mathbb{P}_r$.

Training a neural network as a critic was first proposed in the context of generative modeling in [12]. The particular choice of loss functional has been introduced in [5] to train a critic that captures the Wasserstein distance between the distributions $\mathbb{P}_r$ and $\mathbb{P}_n$. A minimizer to (6) approximates a maximizer $f$ to the Kantorovich formulation of optimal transport [26].

$$\text{Wass}(\mathbb{P}_r, \mathbb{P}_n) = \sup_{f \in 1-Lip} \mathbb{E}_{X \sim \mathbb{P}_n} [f(X)] - \mathbb{E}_{X \sim \mathbb{P}_r} [f(X)]. \tag{7}$$

Relaxing the hard Lipschitz constraint in (7) into a penalty term as in (6) was proposed in [13]. Tracking the gradients of $\Psi_\Theta$ for our experiments demonstrates that this way the Lipschitz constraint can in fact be enforced up to a small error.

---

**Algorithm 1** Learning a regularization functional

---

**Require:** Gradient penalty coefficient $\mu$, batch size $m$, Adam hyperparameters $\alpha$, inverse $A_\delta^+$
  **while** $\Theta$ has not converged **do**
    **for** $i \in 1, ..., m$ **do**
      Sample ground truth image $x_r \sim \mathbb{P}_r$, measurement $y \sim \mathbb{P}_Y$ and random number $\epsilon \sim U[0,1]$
      $x_n \leftarrow A_\delta^+ y$
      $x_i = \epsilon x_r + (1 - \epsilon) x_n$
      $L_i \leftarrow \Psi_\Theta(x_r) - \Psi_\Theta(x_n) + \mu (\|\nabla_{x_i} \Psi_\Theta(x_i)\| - 1)_+^2$
    **end for**
    $\Theta \leftarrow Adam(\nabla_\Theta \sum_{i=1}^m L_i, \alpha)$
  **end while**

---

**Algorithm 2** Applying a learned regularization functional with gradient descent

---

**Require:** Learned regularization functional $\Psi_\Theta$, measurements $y$, regularization weight $\lambda$, step size $\epsilon$, operator $A$, inverse $A_\delta^+$, Stopping criterion $S$

$x \leftarrow A_\delta^+ y$
**while** $S$ not satisfied **do**
   $x \leftarrow x - \epsilon \nabla_x \left[ \|Ax - y\|_2^2 + \lambda \Psi_\Theta(x) \right]$
**end while**
**return** $x$

---

In the proposed algorithm, gradient descent is used to solve (5). As the neural network is in general non-convex, convergence to a global optimum cannot be guaranteed. However, stable convergence to a critical point has been observed in practice. More sophisticated algorithms like momentum methods or a forward-backward splitting of data term and regularization functional can be applied [10].

## 3.2 Distributional Analysis

Here we analyze the impact of the learned regularization functional on the induced image distribution. More precisely, given a noisy image $x$ drawn from $\mathbb{P}_n$, consider the image obtained by performing a step of gradient descent of size $\eta$ over the regularization functional $\Psi_\Theta$

$$g_\eta(x) := x - \eta \cdot \nabla_x \Psi_\Theta(x). \tag{8}$$

This yields a distribution $\mathbb{P}_\eta := (g_\eta)_\# \mathbb{P}_n$ of noisy images that have undergone one step of gradient descent. We show that this distribution is closer in Wasserstein distance to the distribution of ground truth images $\mathbb{P}_r$ than the noisy image distribution $\mathbb{P}_n$. The regularization functional hence introduces the highly desirable incentive to align the distribution of minimizers of the regularization problem (5) with the distribution of ground truth images.

Henceforth, assume the network $\Psi_\Theta$ has been trained to perfection, i.e. that it is a 1-Lipschitz function which achieves the supremum in (7). Furthermore, assume $\Psi_\Theta$ is almost everywhere differentiable with respect to the measure $\mathbb{P}_n$.

**Theorem 1.** *Assume that $\eta \mapsto \mathrm{Wass}(\mathbb{P}_r, \mathbb{P}_\eta)$ admits a left and a right derivative at $\eta = 0$, and that they are equal. Then,*

$$\frac{\mathrm{d}}{\mathrm{d}\eta} \mathrm{Wass}(\mathbb{P}_r, \mathbb{P}_\eta)|_{\eta=0} = -\mathbb{E}_{X \sim \mathbb{P}_n} \left[ \|\nabla_x \Psi_\Theta(X)\|^2 \right].$$

*Proof.* The proof follows [5, Theorem 3]. By an envelope theorem [20, Theorem 1], the existence of the derivative at $\eta = 0$ implies

$$\frac{\mathrm{d}}{\mathrm{d}\eta} \mathrm{Wass}(\mathbb{P}_r, \mathbb{P}_\eta)|_{\eta=0} = \frac{\mathrm{d}}{\mathrm{d}\eta} \mathbb{E}_{X \sim \mathbb{P}_n}[\Psi_\Theta(g_\eta(X))]|_{\eta=0}. \tag{9}$$

On the other hand, for a.e. $x \in X$ one can bound

$$\left| \frac{\mathrm{d}}{\mathrm{d}\eta} \Psi_\Theta(g_\eta(x)) \right| = |\langle \nabla_x \Psi_\Theta(g_\eta(x)), \nabla_x \Psi_\Theta(x) \rangle| \leq \|\nabla_x \Psi_\Theta(g_\eta(x)\| \cdot \|\nabla_x \Psi_\Theta(x)\| \leq 1, \tag{10}$$

for any $\eta \in \mathbb{R}$. Hence, in particular the difference quotient is bounded

$$\left| \frac{1}{\eta} \left[ \Psi_\Theta(g_\eta(x)) - \Psi_\Theta(x) \right] \right| \leq 1 \tag{11}$$

for any $x$ and $\eta$. By dominated convergence, this allows us to conclude

$$\frac{\mathrm{d}}{\mathrm{d}\eta} \mathbb{E}_{X \sim \mathbb{P}_n}[\Psi_\Theta(g_\eta(X))]|_{\eta=0} = \mathbb{E}_{X \sim \mathbb{P}_n} \frac{\mathrm{d}}{\mathrm{d}\eta} [\Psi_\Theta(g_\eta(X))]|_{\eta=0}. \tag{12}$$

Finally,

$$\frac{\mathrm{d}}{\mathrm{d}\eta} [\Psi_\Theta(g_\eta(X))]|_{\eta=0} = -\|\nabla_x \Psi_\Theta(X)\|^2. \tag{13}$$

$\square$

*Remark* 1. Under the weak assumptions in [13, Corollary 1], we have $\|\nabla_x \Psi_\Theta(x)\| = 1$, for $\mathbb{P}_n$ a.e. $x \in X$. This allows to compute the rate of decay of Wasserstein distance explicitly to

$$\frac{\mathrm{d}}{\mathrm{d}\eta}[\Psi_\Theta(g_\eta(X))]|_{\eta=0} = -1 \tag{14}$$

Note that the above calculations also show that the particular choice of loss functional is optimal in terms of decay rates of the Wasserstein distance, introducing the strongest incentive to align the distribution of reconstructions with the ground truth distribution amongst all regularization functionals. To make this more precise, consider any other regularization functional $f : X \to \mathbb{R}$ with norm-bounded gradients, i.e. $\|\nabla f(x)\| \leq 1$.

**Corollary 1.** *Denote by $\tilde{g}_\eta(x) = x - \eta \cdot \nabla f(x)$ the flow associated to $f$. Set $\tilde{\mathbb{P}}_\eta := (\tilde{g}_\eta)_\#(\mathbb{P}_n)$. Then*

$$\frac{\mathrm{d}}{\mathrm{d}\eta} \mathrm{Wass}(\mathbb{P}_r, \tilde{\mathbb{P}}_\eta)|_{\eta=0} \geq -1 = \frac{\mathrm{d}}{\mathrm{d}\eta} \mathrm{Wass}(\mathbb{P}_r, \mathbb{P}_\eta)|_{\eta=0} \tag{15}$$

*Proof.* An analogous computation as above shows

$$\frac{\mathrm{d}}{\mathrm{d}\eta} \mathrm{Wass}(\mathbb{P}_r, \tilde{\mathbb{P}}_\eta)|_{\eta=0} = -\mathbb{E}_{X \sim \mathbb{P}_n} \left[ \langle \nabla_x \Psi_\Theta(x), \nabla_x f(x) \rangle \right] \geq -1 = -\mathbb{E}_{X \sim \mathbb{P}_n} \left[ \|\nabla_x \Psi_\Theta(X)\|^2 \right].$$

$\square$

### 3.3 Analysis under data manifold assumption

Here we discuss which form of regularization functional is desirable under the data manifold assumption and show that the loss function (6) in fact gives rise to a regularization functional of this particular form.

**Assumption 1** (Weak Data Manifold Assumption). *Assume the measure $\mathbb{P}_r$ is supported on the weakly compact set $\mathcal{M}$, i.e. $\mathbb{P}_r(\mathcal{M}^c) = 0$*

This assumption captures the intuition that real data lies in a curved lower-dimensional subspace of $X$.

If we consider the regularization functional as encoding prior knowledge about the image distribution, it follows that we would like the regularizer to penalize images which are away from $\mathcal{M}$. An extreme way of doing this would be to set the regularization functional as the characteristic function of $\mathcal{M}$. However, this choice of functional comes with two major disadvantages: First, solving (5) with methods based on gradient descent becomes impossible when using such a regularization functional. Second, the functional effectively leads to a projection onto the data manifold, possibly causing artifacts due to imperfect knowledge of $\mathcal{M}$ [8].

An alternative to consider is the distance function to the data manifold $d(x, \mathcal{M})$, since such a choice provides meaningful gradients everywhere. This is implicitly done in [21]. In Theorem 2, we show that our chosen loss function in fact does give rise to a regularization functional $\Psi_\Theta$ taking the desirable form of the $l^2$ distance function to $\mathcal{M}$.

Denote by

$$P_\mathcal{M} : D \to \mathcal{M}, \quad x \to \mathrm{argmin}_{y \in \mathcal{M}} \|x - y\| \tag{16}$$

the data manifold projection, where $D$ denotes the set of points for which such a projection exists. We assume $\mathbb{P}_n(D) = 1$. This can be guaranteed under weak assumptions on $\mathcal{M}$ and $\mathbb{P}_n$.

**Assumption 2.** *Assume the measures $\mathbb{P}_r$ and $\mathbb{P}_n$ satisfy*

$$(P_\mathcal{M})_\#(\mathbb{P}_n) = \mathbb{P}_r \tag{17}$$

*i.e. for every measurable set $A \subset X$, we have $\mathbb{P}_n(P_\mathcal{M}^{-1}(A)) = \mathbb{P}_r(A)$*

We hence assume that the distortions of the true data present in the distribution of pseudo-inverses $\mathbb{P}_n$ are well-behaved enough to recover the distribution of true images from noisy ones by projecting back onto the manifold. Note that this is a much weaker than assuming that any given single image can be recovered by projecting its pseudo-inverse back onto the data manifold. Heuristically, Assumption 2 corresponds to a low-noise assumption.

**Theorem 2.** *Under Assumptions 1 and 2, a maximizer to the functional*

$$\sup_{f \in 1-Lip} \mathbb{E}_{X \sim \mathbb{P}_n} f(X) - \mathbb{E}_{X \sim \mathbb{P}_r} f(X) \tag{18}$$

*is given by the distance function to the data manifold*

$$d_{\mathcal{M}}(x) := \min_{y \in \mathcal{M}} \|x - y\| \tag{19}$$

*Proof.* First show that $d_{\mathcal{M}}$ is Lipschitz continuous with Lipschitz constant 1. Let $x_1, x_2 \in X$ be arbitrary and denote by $\tilde{y}$ a minimizer to $\min_{y \in \mathcal{M}} \|x_2 - y\|_2$. Indeed,

$$d_{\mathcal{M}}(x_1) - d_{\mathcal{M}}(x_2) = \min_{y \in \mathcal{M}} \|x_1 - y\| - \min_{y \in \mathcal{M}} \|x_2 - y\| = \min_{y \in \mathcal{M}} \|x_1 - y\| - \|x_2 - \tilde{y}\|$$

$$\leq \|x_1 - \tilde{y}\| - \|x_2 - \tilde{y}\| \leq \|x_1 - x_2\|,$$

where we used the triangle inequality in the last step. This proves Lipschitz continuity by exchanging the roles of $x_1$ and $x_2$.

Now, we prove that $d_{\mathcal{M}}$ obtains the supremum in 18. Let $h$ be any 1-Lipschitz function. By assumption 2, one can rewrite

$$\mathbb{E}_{X \sim \mathbb{P}_n} [h(X)] - \mathbb{E}_{X \sim \mathbb{P}_r} [h(X)] = \mathbb{E}_{X \sim \mathbb{P}_n} [h(X) - h(P_{\mathcal{M}}(X))]. \tag{20}$$

As $h$ is 1 Lipschitz, this can be bounded via

$$\mathbb{E}_{X \sim \mathbb{P}_n} [h(X) - h(P_{\mathcal{M}}(X))] \leq \mathbb{E}_{X \sim \mathbb{P}_n} [\|X - P_{\mathcal{M}}(X)\|]. \tag{21}$$

The distance between $x$ and $P_{\mathcal{M}}(x)$ is by definition given by $d_{\mathcal{M}}(x)$. This allows to conclude via

$$\mathbb{E}_{X \sim \mathbb{P}_n} [\|X - P_{\mathcal{M}}(X)\|] = \mathbb{E}_{X \sim \mathbb{P}_n} [d_{\mathcal{M}}(X)] = \mathbb{E}_{X \sim \mathbb{P}_n} [d_{\mathcal{M}}(X) - d_{\mathcal{M}}(P_{\mathcal{M}}(X))]$$

$$= \mathbb{E}_{X \sim \mathbb{P}_n} [d_{\mathcal{M}}(X)] - \mathbb{E}_{X \sim \mathbb{P}_r} [d_{\mathcal{M}}(X)].$$

$\square$

*Remark* 2 (Non-uniqueness). The functional (18) does not necessarily have a unique maximizer. For example, $f$ can be changed to an arbitrary 1-Lipschitz function outside the convex hull of $\text{supp}(\mathbb{P}_r) \cap \text{supp}(\mathbb{P}_n)$.

## 4    Stability

Following the well-developed stability theory for classical variational problems [11], we derive a stability estimate for the adversarial regularizer algorithm. The key difference to existing theory is that we do not assume the regularization functional $f$ is bounded from below. Instead, this is replaced by a 1 Lipschitz assumption on $f$.

**Theorem 3** (Weak Stability in Data Term). *We make Assumption 3. Let $y_n$ be a sequence in $Y$ with $y_n \to y$ in the norm topology and denote by $x_n$ a sequence of minimizers of the functional*

$$\text{argmin}_{x \in X} \|Ax - y_n\|^2 + \lambda f(x)$$

*Then $x_n$ has a weakly convergent subsequence and the limit $x$ is a minimizer of $\|Ax - y\|^2 + \lambda f(x)$.*

The assumptions and the proof are contained in Appendix A.

## 5    Computational Results

### 5.1    Parameter estimation

Applying the algorithm to new data requires choosing a regularization parameter $\lambda$. Making the assumption that the ground truth images are critical points of the variational problem (5), $\lambda$ can be estimated efficiently from the noise level, using the fact that the regularization functional has gradients of unit norm. This leads to the formula

$$\lambda = 2 \, \mathbb{E}_{e \sim p_n} \|A^* e\|_2,$$

where $A^*$ denotes the adjoint and $p_n$ the noise distribution. In all experiments, the regularization parameter has been chosen according to this formula without further tuning.

Table 1: Denoising results on BSDS dataset

| Method | PSNR (dB) | SSIM |
|---|---|---|
| Noisy Image | 20.3 | .534 |
| MODEL-BASED | | |
| Total Variation [23] | 26.3 | .836 |
| SUPERVISED | | |
| Denoising N.N. [28] | 28.8 | .908 |
| UNSUPERVISED | | |
| Adversarial Regularizer (ours) | 28.2 | .892 |



(a) Ground Truth     (b) Noisy Image     (c) TV     (d) Denoising N.N.   (e) Adversarial Reg.

Figure 1: Denoising Results on BSDS

## 5.2 Denoising

As a toy problem, we compare the performance of total variation denoising [23], a supervised denoising neural network approach [28] based on the UNet [22] architecture and our proposed algorithm on images of size $128 \times 128$ cut out of images taken from the BSDS500 dataset [3]. The images have been corrupted with Gaussian white noise. We report the average peak signal-to-noise ratio (PSNR) and the structural similarity index (SSIM) [27] in Table 1.

The results in Figure 1 show that the adversarial regularizer algorithm is able to outperform classical variational methods in all quality measures. It achieves results of comparable visual quality than supervised data-driven algorithms, without relying on supervised training data.

## 5.3 Computed Tomography

Computer Tomography reconstruction is an application in which the variational approach is very widely used in practice. Here, it serves as a prototype inverse problem with non-trivial forward operator. We compare the performance of total variation [18, 23], post-processing [15], Regularization by Denoising (RED) [21] and our proposed regularizers on the LIDC/IDRI database [6] of lung scans. The denoising algorithm underlying RED has been chosen to be the denoising neural network previously trained for post-processing. Measurements have been simulated by taking the ray transform, corrupted with Gaussian white noise. With 30 different angles taken for the ray transform, the forward operator is undersampled. The code is available online [1].

The results on different noise levels can be found in Table 2 and Figure 2, with further examples in Appendix C. Note in Table 2 that Post-Processing has been trained with PSNR as target loss function. Again, total variation is outperformed by a large margin in all categories. Our reconstructions are of the same or superior visual quality than the ones obtained with supervised machine learning methods, despite having used unsupervised data only.

## 6 Conclusion

We have proposed an algorithm for solving inverse problems, using a neural network as regularization functional. We have introduced a novel training algorithm for regularization functionals and showed that the resulting regularizers have desirable theoretical properties. Unlike other data-based

Table 2: CT reconstruction on LIDC dataset

(a) High noise

| Method | PSNR (dB) | SSIM |
|---|---|---|
| MODEL-BASED | | |
| Filtered Backprojection | 14.9 | .227 |
| Total Variation [18] | 27.7 | .890 |
| SUPERVISED | | |
| Post-Processing [15] | 31.2 | .936 |
| RED [21] | 29.9 | .904 |
| UNSUPERVISED | | |
| Adversarial Reg. (ours) | 30.5 | .927 |

(b) Low noise

| Method | PSNR (dB) | SSIM |
|---|---|---|
| MODEL-BASED | | |
| Filtered Backprojection | 23.3 | .604 |
| Total Variation [18] | 30.0 | .924 |
| SUPERVISED | | |
| Post-Processing [15] | 33.6 | .955 |
| RED [21] | 32.8 | .947 |
| UNSUPERVISED | | |
| Adversarial Reg. (ours) | 32.5 | .946 |

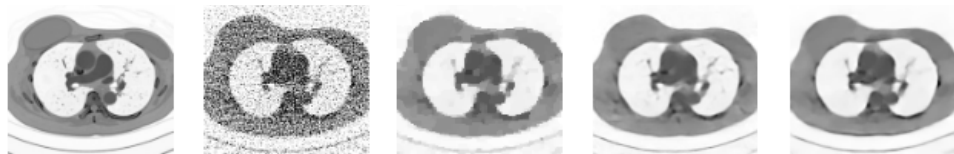

(a) Ground Truth    (b) FBP    (c) TV    (d) Post-Processing (e) Adversarial Reg.

Figure 2: Reconstruction from simulated CT measurements on the LIDC dataset

approaches in inverse problems, the proposed algorithm can be trained even if only unsupervised training data is available. This allows to apply the algorithm to situations where -due to a lack of appropriate training data- machine learning methods have not been used yet.

The variational framework enables us to effectively insert knowledge about the forward operator and the noise model into the reconstruction, allowing the algorithm to be trained on little training data. It also comes with the advantages of a well-developed stability theory and the possibility of adapting the algorithms to different noise levels by changing the regularization parameter $\lambda$, without having to retrain the model from scratch.

The computational results demonstrate the potential of the algorithm, producing reconstructions of the same or even superior visual quality as the ones obtained with supervised approaches on the LIDC dataset, despite the fact that only unsupervised data has been used for training. Classical methods like total variation are outperformed by a large margin.

Our approach is particularly well-suited for applications in medical imaging, where usually very few training samples are available and ground truth images to a particular measurement are hard to obtain, making supervised algorithms impossible train.

## 7    Extensions

The algorithm admits some extensions and modifications.

- *Local* Regularizers. The regularizer is restricted to act on small patches of pixels only, giving the value of the regularization functional by averaging over all patches. This allows to harvest many training samples from a single image, making the algorithm trainable on even less training data. Local Adversarial Regularizers can be implemented by choosing a neural network architecture consisting of convolutional layers followed by a global average pooling.

- Recursive Training. When applying the regularization functional, the variational problem has to be solved. In this process, the regularization functional is confronted with partially reconstructed images, which are neither ground truth images nor exhibit the typical noise distribution the regularization functional has been trained on. By adding these images to the

samples the regularization functional is trained on, the neural network is enabled to learn from its own outputs. First implementations show that this can lead to an additional boost in performance, but that the choice of which images to add is very delicate.

## 8    Acknowledgments

We thank Sam Power, Robert Tovey, Matthew Thorpe, Jonas Adler, Erich Kobler, Jo Schlemper, Christoph Kehle and Moritz Scham for helpful discussions and advice.

The authors acknowledge the National Cancer Institute and the Foundation for the National Institutes of Health, and their critical role in the creation of the free publicly available LIDC/IDRI Database used in this study. The work by Sebastian Lunz was supported by the EPSRC grant EP/L016516/1 for the University of Cambridge Centre for Doctoral Training, the Cambridge Centre for Analysis and by the Cantab Capital Institute for the Mathematics of Information. The work by Ozan Öktem was supported by the Swedish Foundation for Strategic Research grant AM13-0049. Carola-Bibiane Schönlieb acknowledges support from the Leverhulme Trust project on 'Breaking the non-convexity barrier', EPSRC grant Nr. EP/M00483X/1, the EPSRC Centre Nr. EP/N014588/1, the RISE projects CHiPS and NoMADS, the Cantab Capital Institute for the Mathematics of Information and the Alan Turing Institute.

## Footnotes

[1] https://github.com/lunz-s/DeepAdverserialRegulariser

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
