[Supplementary Material]

# Appendix

## A Stability Theory

**Assumption 3.** *All of the following three conditions hold true.*

- *$f$ is lower semi-continuous with respect to the weak topology on $X$ and $1$ Lipschitz with respect to the metric induced by the norm.*

- *$A$ is continuous or equivalently weak-to-weak continuous*

- *One of the following two conditions hold true*

  - *$0 < \alpha := \inf_x \frac{\|Ax\|}{\|x\|}$*
  - *$\|f(x)\| \to \infty$ as $\|x\| \to \infty$*

These assumptions are standard in the classical stability theory for inverse problems [11], with the difference that we assume $f$ to be $1$ Lipschitz instead of being bounded from below.

Next we show that in the particular setting $S = d_{\mathcal{M}}$ where $d_{\mathcal{M}}$ is the distance function to the manifold $\mathcal{M}$, the weak continuity assumption on is always satisfied.

**Lemma 1.** *The map $d_{\mathcal{M}}$ is weakly lower semi-continuous.*

*Proof.* Let $x_n$ be a sequence in $X$ with $x_n \to x$ weakly. Pick any subsequence of $x_n$, for convenience still denoted by $x_n$. Denote by $y_n$ elements in $\mathcal{M}$ such that

$$d_{\mathcal{M}}(x_n) = \min_{y \in \mathcal{M}} \|x_n - y\| = \|x_n - y_n\|.$$

As all $y_n \in \mathcal{M}$, we can extract a weakly convergent subsequence, denoted by $y_{n_j}$, such that $y_{n_j} \to \overline{y}$ weakly for some $\overline{y} \in \mathcal{M}$. By lower semi-continuity of the norm, estimate

$$\liminf_{j \to \infty} d_{\mathcal{M}}(x_{n_j}) = \liminf_{j \to \infty} \|x_{n_j} - y_{n_j}\| \geq \|x - \overline{y}\| \geq \min_{y \in \mathcal{M}} \|x - y\| = d_{\mathcal{M}}(x) \qquad (22)$$

$\square$

**Lemma 2** (Coercivity). *Let $y \in Y$. Then under assumptions 3,*

$$\|Ax - y\|^2 + \lambda f(x) \to \infty$$

*as $\|x\| \to \infty$, uniformly in all $y \in Y$ with $\|y\| \leq 1$.*

*Proof.* Assume first $0 < \alpha := \inf_x \frac{\|Ax\|}{\|x\|}$. WLOG assume $\|x\| \geq \alpha^{-1}$. Then

$$
\begin{aligned}
\|Ax - y\|^2 + \lambda f(x) &\geq (\alpha\|x\| - 1)^2 + \lambda(f(x) - f(0) + f(0)) \\
&\geq (\alpha\|x\| - 1)^2 - \lambda\|x\| + f(0) \to \infty
\end{aligned}
$$

as $\|x\| \to \infty$, uniformly in $y$ with $\|y\| \leq 1$. The last inequality uses the assumption that $f$ is $1$ Lipschitz.

In the case $\|S(x)\| \to \infty$ as $\|x\| \to \infty$ the statement follows immediately. $\square$

**Theorem 4** (Existence of Minimizer). *Under assumptions 3, there exists a minimizer of*

$$\|Ax - y\|^2 + \lambda f(x).$$

*Proof.* Let $x_n$ be a sequence in $X$ such that

$$\|Ax_n - y\|^2 + \lambda f(x_n) \to \min_{x \in X} \|Ax - y\|^2 + \lambda f(x)$$

as $n \to \infty$. Then by Lemma 2, $x_n$ is bounded in norm allowing to extract a weakly convergent subsequence $x_n \to x$. As the norm is weakly lower-semi continuous and so is $f$ by assumption, we obtain

$$\min_{x \in X} \|Ax - y\|^2 + \lambda f(x) = \liminf_{n \to \infty} \|Ax_n - y\|^2 + \lambda f(x_n) \geq \|Ax - y\|^2 + \lambda f(x)$$

thus proving that $x$ is indeed a minimizer. $\square$

**Theorem 5** (Weak Stability in Data Term). *Assume 3. Let $y_n$ be a sequence in $Y$ with $y_n \to y$ in the norm topology and denote by $x_n$ a sequence of minimizers of the functional*

$$\mathrm{argmin}_{x \in X} \|Ax - y_n\|^2 + \lambda f(x)$$

*Then $x_n$ has a weakly convergent subsequence and the limit $x$ is a minimizer of $\|Ax - y\|^2 + \lambda f(x)$.*

*Proof.* By Lemma 2, the sequence $\|x_n\|$ is bounded and hence contains a weakly convergent subsequence $x_n \to x$. Note that the map $L(y) = \min_x \|Ax - y\|^2 + \lambda f(x)$ is continuous with respect to the norm topology. On the other hand, as $A$ is by assumption weak to weak continuous and both the norm and $f$ are weakly lower semi-continuous, the overall loss $L(x, y) = \|Ax - y\|^2 + \lambda f(x)$ is weakly lower semi-continuous, so

$$\liminf_{n \to \infty} L(x_n, y_n) \geq L(x, y).$$

Together we obtain

$$L(y) = \lim_{k \to \infty} L(y_k) = \lim_{k \to \infty} L(x_k, y_k) \geq L(x, y), \tag{23}$$

proving that the limit point $x$ is indeed a minimizer of $\|Ax - y\|^2 + \lambda f(x)$. $\qquad\square$

## B Implementation details

We used a simple 8 layer convolutional neural network with a total of four strided convolution layers with stride 2, leaky ReLU ($\alpha = 0.1$) activations and two final dense layer for all experiments with the adversarial regularizer algorithm. The network was optimized with RMSProp. We solved the variational problem using gradient descent with fixed step size. The regularization parameter was chosen according to the heuristic given in paper.

The comparison experiments with Post-Processing and the Denoising Neural Network used a UNet style architecture, with four down-sampling (strided convolution, stride 2) and four up-sampling (transposed convolution) convolutional layers with skip-connections after every down-sampling step to the corresponding up-sampled layer of the same image resolution. Again, leaky ReLU activations were used. The network was optimized using Adam. As training loss we used the $\ell^2$ distance to the ground truth, with no further regularization terms on the network parameters.

In the experiments with total variation, the regularization parameter was chosen using line search, picking the parameter that leads to the best PSNR value. The minimization problem was solved using primal-dual hybrid gradient descent (PDHG) [10].

## C  Further Computational Results

Figure 3: Further denoising results on BSDS dataset.
From left to right: Ground truth, Noisy Image, TV, Denoising Neural Network, Adversarial Reg.

Figure 4: Further CT reconstructions on LIDC dataset, high noise.
From left to right: Ground truth, FBP, TV, Post-Processing, Adversarial Reg.

(a) Ground Truth          (b) Total Variation          (c) Adversarial Regularizer

Figure 5: Adversarial Regularizers cause fewer artifacts around-small angle intersections of different domains than TV. Results obtained for CT Reconstruction on synthetic ellipse data.

Figure 6: Further CT reconstructions on LIDC dataset, low noise.
From left to right: Ground truth, FBP, TV, Post-Processing, Adversarial Reg.
Below the Sinogram used for reconstruction of the images.