[Reviews · NeurIPS 2018]

Reviewer 1



Summary: The paper proposes to learn the regularization function in inverse problem. This idea is that the regularization should take large values for non-plausible images and low values (not necessaritly bounded below) for true images. While many variational formulations use hard coded functionals such as TV for images, here it is learnt using an approach inspired by the Wasserstein GAN method. To learn the regularization the GAN has to discrimitate between measurements backprojected to image space using a simple pseudo-inverse strongly affected by noise, and some ground truth images. By promoting a regularization function which is "close" to have 1-Lipschitz gradient, results on stability can be obtained and hyperparameter \lambda can be calibrated using the same network that does not depend on the noise level. The paper is well written. Qualitative evaluation: - While the GAN promotes a regularizer with a 1-Lipschitz gradient it is not a hard constraint of the learning problem. However theoretical results make the assumption that the function has a 1-Lipschitz gradient everywhere. This weakens the theoretical claims of the paper, if not makes the theorems innapropriate. This should be discussed. - One strong claim of the paper is that the method does not need lots of ground truth data. "Our approach is particularly well-suited for applications in medical imaging, where usually very few training samples are available and ground truth images to a particular measurement are hard to obtain, making supervised algorithms impossible train."" However the procedure described in Algorithm 1 requires to simulate data using ground truth images. This seems to be a paradox and there is no experimental evidence that shows that the method needs very little ground truth data to perform well. Misc / Typos: - Careful with capital letters in the titles of the papers in the bibliography. - Kontorovich -> Kantorovich

Reviewer 2



SUMMARY: The authors propose to learn a representation of regularization functionals in inverse reconstruction problems by following adversarial training approach. Theoretical derivations of the problem and its stability are presented, illustrative cases implemented for demonstration purposes. PRO: * the overall idea of learning a problem (patient?) specific reconstruction function, or regularization functional, is interesting. * the authors presented a dedicated theoretic treatment of the matter CON * results are little more than case studies, the chosen examples (at the end and in the supplementary material) demonstrate only limited advantages in qualitative and quantitative terms. * I somewhat feel that "Adversarial Regularizers in Inverse Problems" might oversell the purpose of solving a PET/CT image reconstruction task.

Reviewer 3



This paper proposes a framework for training an adversarial discriminator, and using it as a prior for signal reconstruction problems. The authors ask theoretical questions about when such a regularizer would be successful. The theoretical questions are well thought out, and the proofs are fairly simple and do not reveal any major new analysis methods. The experimental setting is well setup (although lacking a few things I'd like to see) and I appreciate that the authors were upfront about the fact that their algorithm doesn't achieve the best performance on the denoising task. While I think this paper is worth accepting, I have some qualms because the experiments feel a bit incomplete. The are of regularization using neural nets is becoming somewhat mature, and a range of methods exist, including plug-and-play, and "regularization by denoising."